# Age-Period-Cohort Study of Breast Cancer Mortality in Brazil in State Capitals and in Non-Capital Municipalities from 1980 to 2019

**DOI:** 10.3390/ijerph20156505

**Published:** 2023-08-02

**Authors:** Rodrigo Chávez-Penha, Maria Teresa Bustamante-Teixeira, Mário Círio Nogueira

**Affiliations:** Post-Graduation Program in Collective Health, Faculty of Medicine, Federal University of Juiz de Fora, Juiz de Fora 36036-900, Minas Gerais, Brazil; mariateresa.bustamante@ufjf.br

**Keywords:** breast cancer, mortality, age-period-cohort model

## Abstract

Breast cancer was identified as the cancer with the highest mortality rate among women in Brazil. This study analyzed the effects of age, period and birth cohort on the breast cancer mortality rate for Brazilian women, comparing state capitals and non-capital municipalities. Population and deaths data were extracted from the Brazilian Unified Health System database for women aged 30 years or older, for the years between 1980 and 2019. The effects were analyzed using the age-period-cohort model. Age effect on breast cancer mortality is observed in the model through higher mortality rates at older ages. Period effect is similar in all regions in the form of a marked increase in the rate ratio (RR) in non-capital municipalities by period than in state capitals. The RR of birth cohorts in the state capitals remained stable (north, northeast and central-west regions) or decreased followed by an increase in the most recent cohorts (Brazil as a whole and the southeast and south regions). The RR for the other municipalities, however, showed a progressive increase in the cohorts for all regions. Policies and actions focused on breast cancer in women should consider these differences among Brazilian regions, state capitals and other municipalities.

## 1. Introduction

In 2020, breast cancer was identified as the most common cancer in the world (excluding nonmelanoma skin tumors) and as the cancer with the highest mortality rate among women [1,2]. In Brazil, the same is observed [3,4], and for decades the increase in cases and deaths from this cancer has drawn the attention of researchers and decision-makers at various levels, with various control policies being adopted at different times [5].

Studies on the breast cancer mortality trend in women therefore seek to capture the effects of these measures as well as to identify in more detail the causes and factors influencing the trend. The age-period-cohort allows researchers to differentiate these elements within the evolution of indicators of health problems to determine the causal effect of each of these processes in changes observed over time [6,7]. These elements can, be described as age effects, such as variations associated with aging, period effects, such as variations associated with events that occurred in calendar-based time in which an outcome of interest occurs and birth cohort effects, such as variations associated with the exposure of a group born at the same time to events that occur at various stages of life, especially those that may have future effects [8].

The study of these factors has been used to analyze mortality from breast cancer in several countries around the world, contributing to a better understanding of the relationship between these effects and breast cancer mortality [9,10,11,12,13,14]. In Brazil, this methodology has been previously applied in analyses by region [15], leaving room for more detailed studies. Thus, to contribute to a better understanding and explore possible inequalities in Brazil, the aim of this study was to analyze the effects of age, period and birth cohort on the breast cancer mortality rate for Brazilian women, comparing state capitals and non-capital municipalities.

## 2. Materials and Methods

Population data were extracted from a database of the Department of Informatics of the Brazilian Unified Health System (DATASUS—https://datasus.saude.gov.br/, accessed on 16 July 2022) for women only, in groups of 5 years beginning with 30 years of age. For the years between 1980 and 2009, data from the censuses (1980, 1991, 2000 and 2010), count (1996) and intercensal projections (1981 to 2012) were used. For the years between 2010 and 2019, the study of population estimates by municipality, age and sex from 2000 to 2020 were used.

Data on deaths were also obtained from the DATASUS database. Deaths of women due to breast cancer that occurred between 1980 and 1995 were extracted using code 174 of the 9th version of the International Classification of Diseases (ICD-9). For the years between 1996 and 2019, the code C50 of the 10th version (ICD-10) was used. Only deaths among women were selected, classified by age group and place of residence. The data were grouped into 5-year age groups, from 30 years of age to 80 years of age or older.

The underreporting of breast cancer deaths was corrected considering deaths from ill-defined causes using the methodology proposed by the World Health Organization (WHO) [16], as exemplified by Couto et al. [17]. This method recommends the redistribution of ill-defined causes by a two-step procedure. First, a percentage of adjustment for ill-defined causes (PAIDCs) is calculated for each municipality and age group in each period using total female deaths, female external deaths and ill-defined deaths. Second, from the PAIDC, a correction factor is calculated and then multiplied to the total number of deaths in each municipality in the same period according to each age group (formulas are presented in Appendix A).

From the data obtained, groups were defined for analyses of the effects of age and period. Birth cohorts were calculated by subtracting age from year of death (cohort= period-age), in accordance with the classical method [6], generating data for the missing parameter for building the models. Thus, 11 age groups, 8 periods and 18 birth cohorts were defined.

For the analyses, data from the capitals of the 26 Brazilian states and the Federal District were grouped as state capital data. The information on population and deaths in the states was then subtracted from the values for the state capitals, characterizing the ‘non-capital municipalities’ group. Subgroups were organized by region of Brazil, i.e., north, northeast, central-west, southeast and south, following the same criteria.

The effects were analyzed using the age-period-cohort model proposed by Holford [6] and adapted by Clayton and Schifflers [18] and Carstensen [19]. The choice to use this model is justified by its suitability for the proposed analysis and its use in previous studies, thus enabling comparisons with previous findings. For the analysis of effects, to address the identifiability problem, a zero-sum constraint was applied to the cohort effects in the assessment of the risks of period effects and vice versa. An important parameter for the analysis of the rate ratio (RR) generated by the models is the establishment of a reference period and cohort. The medians of the data for the capital cities of Brazil were chosen as reference for the models for all regions, with the reference period being the five-year period from 2005 to 2009 and the reference cohort being the five-year period from 1945 to 1949. Data were processed using Excel (Version 2212), RStudio (version 2022.12.0), R (version 4.2.2) and the Epi package, version 2.47.

## 3. Results

The evolution of breast cancer mortality rates for women by age group, by birth cohort and by region in Brazil is shown in Figure 1 and Figure 2 (tables with values are available in Appendix B). In general, there are higher mortality rates in the state capitals than in the non-capital municipalities for all regions. The southern and south regions show the highest values in all age groups, through all cohorts and for both regions. Those two regions also show a similar trend among the capital cities where the mortality rates reduce with the evolution of birth cohorts for the age groups, whereas an upward trend is observed for non-capital municipalities.

Another aspect common to all regions is the higher mortality rates for older women, an expected pattern for breast cancer. There are, however, some differences in the regions, between state capitals and non-capital municipalities, especially the change in the slope of the curves, indicating an increase in cases in the most recent cohorts, with notable differences between the state capitals and the non-capital municipalities. A steep slope upward is observed for rates in non-capital cities in north, northeast and central-west regions as cohorts get newer, indicating that women born more recently have higher mortality rates by breast cancer than previous birth cohorts for the same age. With less variation, the capitals of these regions also present higher mortality for more recent birth cohorts.

The data on the fit of the models generated by the age-period-cohort method are presented in Table 1, where possible study models are evaluated considering the inclusion of variables, degrees of freedom, deviance and *p*-value (Chi-square) of each model. For all regions, the models with the three variables (age, period and cohort) show the best fit based on deviance. Only in the north region does the 3-variable model not show statistical significance (*p* > 0.05).

The results for the effects of age, period and cohort are presented in G Figure 3 for Brazil and Brazilian regions, with the curves representing the state capitals and non-capital municipalities, flanked by dashed lines representing the 95% confidence interval. The graphs separately show the effect of age, the effect of periods and the effect of birth cohorts on mortality rates (tables with values are available in the Appendix B).

The effect of age on breast cancer mortality is observed in the model through higher mortality rates at older ages. In this sense, in all regions, mortality rates in state capitals are higher than those in non-capital municipalities. Notably, there were two pronounced increases: one between 30 and 54 years of age and the other after 75 years of age. Between such increases, i.e., in the 55 to 74 age group, rates continue to increase but not as markedly.

The estimated period effect is similar in all regions in the form of a more marked increase in the rate ratio (RR) in non-capital municipalities than in the state capitals. For most regions, the RR in non-capital municipalities exceeds that of the state capitals after the reference period (2005–2009), except for the north and central-west regions, where the RRs of the state capitals and of the non-capital municipalities increase similarly. The values for non-capital municipalities also show, in most regions, a sustained increase in RR, surpassing the value of the 2005–2009 period and continuing to increase until the end of the observed period. The data for the state capitals show different progressions. The RRs of the state capitals of the north and central-west regions show small variations through the whole observed period, remaining close to or at the 1 value until the period of 2010–2014, where an increase trend is observed. The northeast region remained stable during the entire period, not showing variations in the RR from 1. The RRs of Brazil and southeast region capitals start around 1.0 and rise until the 1995–1999 period, where they show a marked reduction after this period. The south region starts with a RR a little higher than 1.1, then remains stable until the 1995–1999 period, followed by a decrease similar to the one observed for the southeast region.

The effects of the birth cohorts represented by the RR in relation to the reference cohort show different behaviors by territory. The RR of birth cohorts in the state capitals remain stable (north, northeast and central-west regions) or decrease followed by an increase in the most recent cohorts (Brazil as a whole and the southeast and south regions). The RR for the other municipalities, however, shows a progressive increase in the cohorts for all regions. The trends range from smaller variations, between approximately 0.1 and 0.5 (Brazil and the southeast and south regions), to large variations, above 2.0 and reaching 7.0 (north, northeast and central-west regions). With different trajectories, an increase in RR is observed for recent cohorts, which indicates that women who were born more recently have had a higher RR than previous generations; however, for state capitals of regions such as the northeast and south, a slight reduction is observed in the last cohort recorded (from 1985 to 1989).

## 4. Discussion

The use of the age-period-cohort (APC) method is relevant in the study of time-related effects on the health of populations. This study observed a strong relationship between mortality rates and age and place of residence, with higher rates found among elderly women and in the state capitals of all regions. The period effect seems to be notable for non-capital municipalities, with an increasing RR. The cohort effect differs among regions, with a stable or decreasing RR in state capitals and increasing values in non-capital municipalities.

The breast cancer mortality trend has been the subject of analysis in several studies during various periods in Brazil by region. Those produced in the last 10 years generally showed an increase in mortality between the 1980s and 2000s followed by stability or a slight increase, with variations among regions. These rates decreased in the southeastern and southern regions and increased in the northern and northeastern regions [3,17,20,21,22,23]. In addition to these findings, the study by Meira et al. [15] used the same APC method applied in this study, and the results indicated higher rates for older age groups and a nonlinear pattern in the rates of birth cohorts, with recent cohorts showing an increased risk for cancer deaths.

Other studies around the world that used this same methodology have interesting comparative aspects. The different trends among the territories draw attention to the possibilities of using this method to highlight the impact of different policies, access to technologies, cultures and lifestyles on breast cancer mortality. Although the age effects show similar trends in the different territories, the period effects and the cohort effects seem to follow different directions. While a reduction in mortality has been observed in more recent cohorts in countries such as France [24], the United States of America [10] and Russia [11], there has been an increase in China [12,14], South Korea and Japan [25], and India, Pakistan and Thailand [13]. These results allow for several analyses, from aspects related to cohort effects, such as the possible relationship between a peak in breast cancer deaths in the 1951 cohort in Taiwan and the use of DDT, an insecticide recognized as carcinogen and widely used in this location [26], to period effects such as the reduction in deaths due to organized screening strategies for early detection by mammography [11,24].

Recently, two published articles have compared urban and rural zones regarding the incidence or mortality by breast cancer. One study on Taiwan investigated the differences between rural and urban regions, trying to analyze through APC method the possible interference of westernized habits in breast cancer incidence. They observed that there is a disparity in cancer rates between these two areas, with higher rates in urban areas but a faster increase in rates in rural areas. Additionally, it was observed that the difference gradually disappeared across the cohorts, which was explained by the westernization in similar periods. Other differences were found and concern early-stage diagnosis, which was better in urban regions indicating a probable worse access to health services in rural areas [27]. Another publication studied China and analyzed breast cancer mortality trends in urban and rural areas using the APC method and modeled trends up to 2039 using the Norpred project. In its main results, it is also observed that urban areas show higher mortality rates than rural ones. Despite that, the effects for age and cohort were similar for both regions. While the observed age effect was to increase mortality as people get older, cohort effect showed a downward trend, with recent cohorts with lower risks than initial ones in both regions. The period effect presented a curious ‘M’ shaped pattern, very similar to both areas, but ultimately with higher effect estimate to urban regions [28]. Similar to this research, these two studies emphasize the importance of understanding differences between areas from the same country to better comprehend country level trends as a reflection of where most of the events of interest are happening, and not necessarily where most help or focus is needed. 

The effect of age on breast cancer mortality on women found in this study is epidemiologically well known; however, its mechanisms are the subject of scientific research to better understand its process. First, it is important to highlight some changes observed in Brazil in the studied period: the demographic transition [29] (the proportional increase in the number of elderly people in the Brazilian population), the epidemiological transition [30] (the change in the main causes of mortality from infectious diseases to chronic noncommunicable diseases) and the increase in life expectancy of Brazilian women, which increased by 31.8 years, from 48.3 to 80.1 years, from 1940 to 2019 [31]. The increased mortality from breast cancer with aging seems to reflect the effect of prolonged exposure as well as other effects of aging, such as frailty in elderly women, chronicity of inflammatory processes and reduced immunity, all aspects generally associated with tumors with slower progression [32,33,34,35].

An aspect related to age but better detectable as a cohort effect is the abrupt recent increase in mortality from breast cancer in younger women between 30 and 50 years of age. This effect is detected in studies in other countries, either as a cohort effect or as an aspect related to younger age groups, also characterized as premenopausal groups [36,37]. One explanation proposed for this result is that changes in the formation of the breast epithelium would induce cancer development early in young women, a group normally not included in mammogram screening programs. These changes need more research to understand if it is related to environmental exposures in early stages of the human development, as in pregnancy, or if they explain the relationship between the use of contemporary hormonal contraceptive methods and breast cancer in young women [38,39]. This could explain the specific increase in high-grade cancers for these age groups, alongside with better diagnosis machines and early diagnosis programs, as in people with known hereditary higher cancer risk, both period effects [40,41,42].

In general, period effects such as new recommendations based on new evidence changes the mortality rate of breast cancer in all age groups, not just the youngest cohort, even if it might be influenced by change for a longer period. Studies such as the National Health Survey (PNS—https://www.pns.icict.fiocruz.br/, accessed on 10 June 2023) and the study of Sustainable Development Indicators (SDIs—https://sidra.ibge.gov.br/pesquisa/ids/tabelas, accessed on 11 June 2023) reveal behavioral changes in Brazil that have been underway for years, such as a reduction in the fertility rate and an increase in pregnancy rates at older ages, changes in diets with higher consumption of ultra-processed foods, and an increase in obesity in women, which are all known risk factors for breast cancer. Cancer is a late result of behavioral changes, since the time from the mutation of a single or small group of cells may take decades to develop into a diagnosable disease. If we consider changes in the other direction, protective and prevention modifications cannot halt or reverse the carcinogenic process.

In that sense, it should be highlighted that the same database shows the increase in habits that reduce the risk for breast cancer. The identified protective changes recognized include reduced smoking, increased leisure-time physical activity and increased educational levels, which come together with other data about increased rates of breastfeeding [43,44,45,46]. In years to come, those changes must be stimulated by health policies that should aim toward breast cancer prevention alongside the enhancement of current early diagnosis strategies.

Period effects and mortality results found in the analyzed periods corroborates those of other studies [15,47] and emphasize the differentiated increase in non-capital municipalities of Brazil [3]. These effects demonstrate that the impacts of government actions and events during the study period affected regions, state capitals and non-capital municipalities differently. There have been well-known initiatives to increase the early diagnosis of cancer since the mid-1980s onward, with subsequent expansion such as the Integrated Women’s Health Care Program (PAISM) and PRO-ONCO, which partially explain the reduction in mortality especially in state capitals in the 1990s, preceding the implementation of mammography as a strategy for the early detection of breast cancer [5,48]. Despite the positive effects of these policies, some articles highlight issues regarding access to mammograms, either due to unequal distribution of machines or due to differentiated access by socioeconomic class [49,50,51,52], which may contribute to the difference in mortality observed in this study. On the other hand, these women with faster access to new technologies were more exposed to prescriptions of hormone replacement therapy (HRT), used in many forms since 1960s up to the 2000s for symptomatic postmenopausal women [53]. That all changed with the Women’s Health Initiative (WHI) studies in 2002 and 2004, with the discovery that those treatments increase the risk of breast cancer, leading to the establishment of stricter criteria for its indication [54].

Another aspect related to period effects is the changing socioeconomic status (SES) of Brazil, which happened differently among regions and affected incidence and mortality by breast cancer. The association between breast cancer and SES is well established [55,56]. Studies in Brazil identified associations of higher SES with higher breast cancer incidence and mortality, especially in northeast region, indicating reasons such as earlier menarche, pregnancies in older ages, lower fertility rates, less time breast feeding, faster demographic transition (increase in elder people proportion of population), changes in diet (with an increase in fat and ultra-processed foods) and health system conditions, i.e., better cancer-specific health services and higher density of doctors providing more breast cancer diagnoses and consequently better definition on death certificates [49,57,58].

In addition to the limitations mentioned about the APC method, another limitation of this study is the coverage and quality of mortality data. These varied in quality by year and region, with problems ranging from the diagnosis of the causes of death to the completion of the declarations, causing underreported deaths or deaths identified as due to ill-defined causes [59,60]. North and northern regions present the most problems, with death underreported. This interferes with the analyses due to possible underestimation of breast cancer deaths, especially in early periods and older cohorts. The World Health Organization method for the correction of the number of deaths by ill-defined causes used in this research offers some improvement in numbers yet remains limited. The effort to apply complex techniques in this adjustment must rely on choices of each researcher to make estimations on how far the data used are from the real numbers. It is worth noting that the quality of records for cancer deaths are better than those for death from other causes [61]. Moreover, as the quality of death records improve findings from research of recent periods tend to be more reliable.

## 5. Conclusions

This study showed that although the highest mortality rates are found in Brazilian state capitals and in the southeast and south regions, the trend of increased mortality is more marked for women born more recently, in the north and northeast regions and, in general, in non-capital municipalities in all regions of the country. Policies and actions on women’s health will have to consider the differences among Brazilian regions, state capitals and other municipalities and the complex diversity of contexts in which the population lives and receives health care. Furthermore, it is necessary to understand Brazil better through studies that investigate possible causes and interventions capable of addressing this complex issue, with the objective of preventing premature death from breast cancer, which has the highest incidence and mortality rate for women in the world today.

## Figures and Tables

**Figure 1 ijerph-20-06505-f001:**
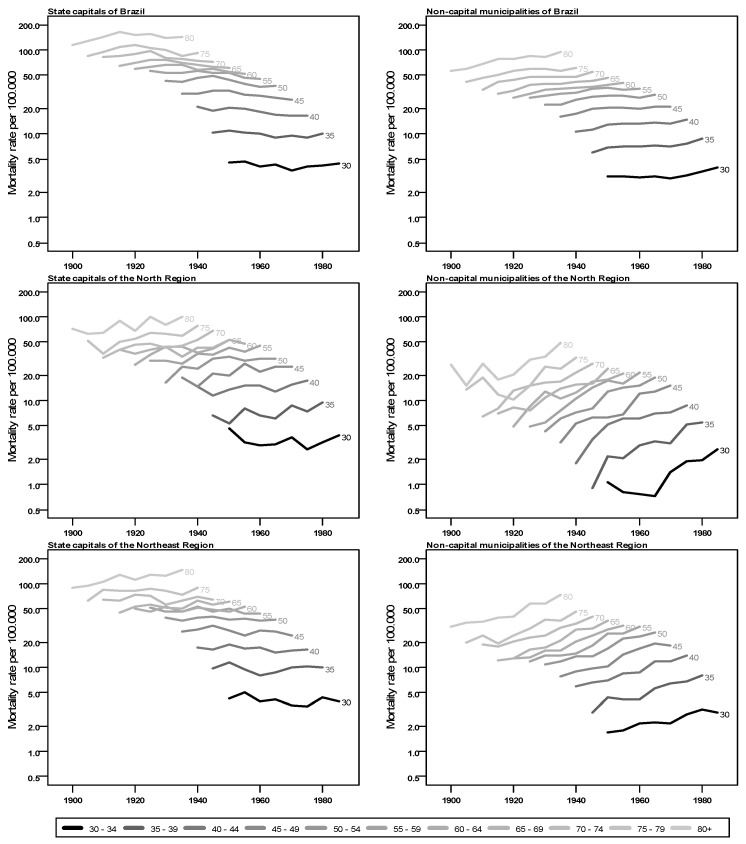
Mortality rates from breast cancer for women (per 100,000) by birth cohort, according to age group and residence in state capitals or in non-capital municipalities of Brazil and of north and northeast Brazilian regions, 1900 to 1989.

**Figure 2 ijerph-20-06505-f002:**
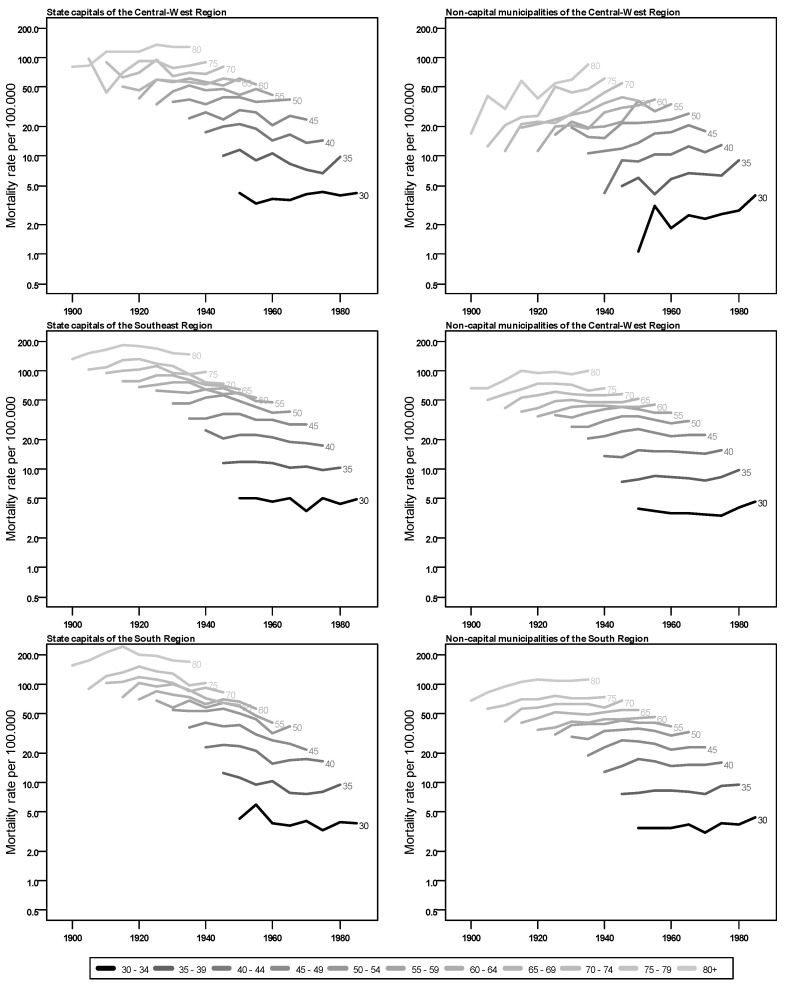
Mortality rates from breast cancer for women (per 100,000) by birth cohort, according to age group and residence in state capitals or in non-capital municipalities of central-west, southeast and south Brazilian regions, 1900 to 1989.

**Figure 3 ijerph-20-06505-f003:**
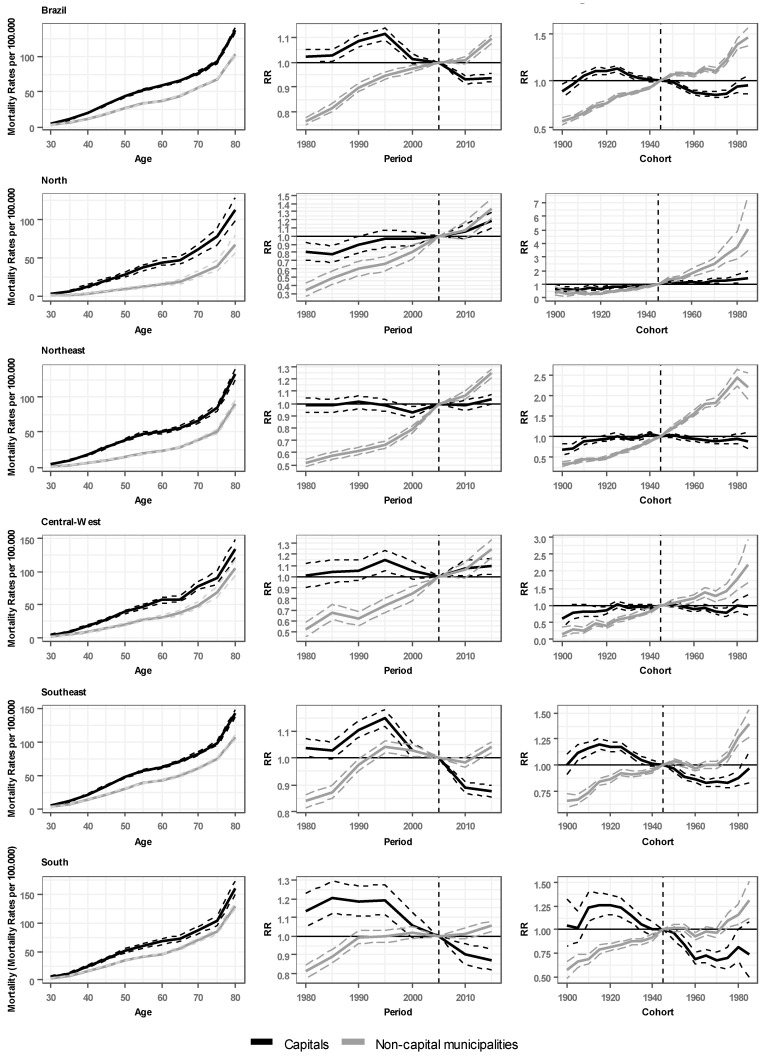
Age-period-cohort model of mortality from breast cancer for women residing in the state capitals and non-capital municipalities of Brazil and Brazilian regions, 1980 to 2019.

**Table 1 ijerph-20-06505-t001:** Fit for the age-period-cohort models for breast cancer mortality in women residing in the state capitals and non-capital municipalities of Brazil and Brazilian regions, 1980 to 2019.

	Brazil						North					
	State Capitals			Non-Capital Municipalities			State Capitals			Non-Capital Municipalities		
**Models**	DF	D	*p*	DF	D	*p*	DF	D	*p*	DF	D	*p*
**Age**	77	1010.18	NA	77	2552.07	NA	77	162.85	NA	77	661.11	NA
**Age-drift**	76	633.77	<0.001	76	607.65	<0.001	76	76.86	<0.001	76	77.55	<0.001
**Age-cohort**	60	298.14	<0.001	60	238.34	<0.001	60	56.40	0.200	60	57.99	0.241
**Age-period-cohort**	54	90.10	<0.001	54	95.18	<0.001	54	49.63	0.343	54	48.73	0.159
**Age-period**	70	318.51	<0.001	70	388.24	<0.001	70	70.35	0.190	70	69.48	0.188
**Age-drift**	76	633.77	<0.001	76	607.65	<0.001	76	76.86	0.369	76	77.55	0.233
	Northeast						Central-West					
	State capitals			Non-capital municipalities			State capitals			Non-capital municipalities		
**Models**	DF	D	*p*	DF	D	*p*	DF	D	*p*	DF	D	*p*
**Age**	77	190.47	NA	77	3982.96	NA	77	120.74	NA	77	620.24	NA
**Age-drift**	76	188.82	0.199	76	221.80	<0.01	76	120.70	0.837	76	168.53	<0.001
**Age-pohort**	60	112.41	<0.001	60	139.25	<0.01	60	73.53	<0.001	60	101.96	<0.001
**Age-period-cohort**	54	91.22	0.002	54	65.86	<0.01	54	59.46	0.029	54	87.06	0.021
**Age-period**	70	170.18	<0.001	70	138.73	<0.01	70	105.40	<0.001	70	153.23	<0.001
**Age-drift**	76	188.82	0.005	76	221.80	<0.01	76	120.70	0.018	76	168.53	0.018
	Southeast						South					
	State capitals			Non-capital municipalities			State capitals			Non-capital municipalities		
**Models**	DF	D	*p*	DF	D	*p*	DF	D	*p*	DF	D	*p*
**Age**	77	1051.56	NA	77	787.98	NA	77	395.09	NA	77	401.58	NA
**Age-drift**	76	547.17	<0.001	76	632.94	<0.001	76	191.51	<0.001	76	279.79	<0.001
**Age-cohort**	60	347.75	<0.001	60	305.18	<0.001	60	86.46	<0.001	60	105.78	<0.001
**Age-period-cohort**	54	87.75	<0.001	54	87.97	<0.001	54	54.97	<0.001	54	40.96	<0.001
**Age-period**	70	202.80	<0.001	70	327.25	<0.001	70	134.02	<0.001	70	175.84	<0.001
**Age-drift**	76	547.17	<0.001	76	632.94	<0.001	76	191.51	<0.001	76	279.79	<0.001

DF: Degrees of freedom; D: Residual deviance; *p*: Chi-square *p*-value.

## Data Availability

Publicly available datasets were analyzed in this study. This data can be found here: https://datasus.saude.gov.br/informacoes-de-saude-tabnet/ (accessed on 16 July 2022).

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
