# Peer review of "Age-Period-Cohort Study of Breast Cancer Mortality in Brazil in State Capitals and in Non-Capital Municipalities from 1980 to 2019"

_ijerph, 2023, doi:10.3390/ijerph20156505_

Round 1
Reviewer 1 Report
The study deals with an extremely relevant topic for Public Health, it seems to have been well conducted and presents important results on breast cancer trends in Brazil and regions.
It presents an additional contribution to the understanding of these trends, which is the assessment of possible cohort effects. In this sense, I believe that the cohort effects identified in the results should be better discussed. Why are capitals and non-capitals so different? Why does the RR increase in non-capital cities? What hypotheses could justify such behavior?
Here are some other questions and suggestions:
In line 32 the authors say: “One method available is age-period-cohort studies, which seek to differentiate these elements in the… “. I believe that a method should not be used because it is available, but because it is the most suitable method to achieve a certain objective. Maybe replace it with: “The age-period-cohort method allows to differentiate these elements in the…”
On line 95: delete “municipalities of the North, Northeast and Central-West regions.”
Line 109: It seems to me that in the Central-West region the best model was the one with 3 variables
Figure 1: Place title on y-axis.
Table 1:
- Adjust the width of rows and columns to improve visualization
- It is necessary to explain which test the p value refers to. I believe it is the test (F or Chi-square?) that compares the model in a given row with the model in the previous row.
Line 130: delete “by period”
Line 132: replace the expression “reference period” with the year in question.
Line 133: This expression is confusing: “surpassing the value of the reference period and continuing to increase until the end of the observed period, which occurs a little later for Brazil “
I believe that the following paragraph could be deleted (line 183), as it presents questions about the age-period-cohort models that have already been discussed in methodological articles and are not part of the questions of this study: “The age-period-cohort method has been used for many years and can provide ... requires the use of techniques such as zero-sum restriction to one of the parameters for each model.”
Line 193 to 200: The text is confusing and unfortunately the example does not help to understand the question.
Line 217: It is not clear how “changes in the formation of the breast epithelium, specific increases in high-grade cancers and late cancer diagnosis” can characterize a cohort effect.
Line 220: The authors say: “In general, the cohort effect indicates changes in the mortality rate from breast cancer in all age groups, in addition to the youngest group, with some that deserve attention”. Changes in the mortality rate in all age groups do not characterize a period effect?
Author Response
Dear reviewer 1,
Thank you for the suggestions. Our answers and comments are highlighted in bold font.
Reviewer 1
Open Review
( ) I would not like to sign my review report
(x) I would like to sign my review report
Quality of English Language
(x) I am not qualified to assess the quality of English in this paper
( ) English very difficult to understand/incomprehensible
( ) Extensive editing of English language required
( ) Moderate editing of English language required
( ) Minor editing of English language required
( ) English language fine. No issues detected
|
Yes |
Can be improved |
Must be improved |
Not applicable |
|
|
Does the introduction provide sufficient background and include all relevant references? |
(x) |
( ) |
( ) |
( ) |
|
Are all the cited references relevant to the research? |
(x) |
( ) |
( ) |
( ) |
|
Is the research design appropriate? |
(x) |
( ) |
( ) |
( ) |
|
Are the methods adequately described? |
(x) |
( ) |
( ) |
( ) |
|
Are the results clearly presented? |
(x) |
( ) |
( ) |
( ) |
|
Are the conclusions supported by the results? |
(x) |
( ) |
( ) |
( ) |
Comments and Suggestions for Authors
The study deals with an extremely relevant topic for Public Health, it seems to have been well conducted and presents important results on breast cancer trends in Brazil and regions.
It presents an additional contribution to the understanding of these trends, which is the assessment of possible cohort effects. In this sense, I believe that the cohort effects identified in the results should be better discussed. Why are capitals and non-capitals so different? Why does the RR increase in non-capital cities? What hypotheses could justify such behavior?
There are indeed some explanations. I’d like to apologize not to present some of them before. This was related to text size choices, since the required number of words were once interpreted as the maximum number of words, and the choice of presenting them separately as age, period or cohort effects. The hypotheses for the differences found in the submitted paper are (hopefully) better presented below and will be added to the manuscript’s new version.
Brazilian capitals have proportionally more diagnosis and treatment resources than non-capital municipalities. This is also true concerning breast cancer care. Nogueira et al (2019), have shown geographical inequities in mammographic screening access in Brazil, correlating it with “high household income inequality, low number of radiologists/100,000 inhabitants, low number of mammography machines/10,000 inhabitants, and low number of mammograms performed by each machine as independent correlates of poor mammographic coverage”, all aspects more present in non-capital municipalities, that include the more rural and poorer geographic regions in the country[1].
The access to mammograms and their productivity may also play a role in the differences found in our paper. Analyzing data from 2016, Rodrigues et al (2019) identified that an insufficient number of mammograms are performed in Brazil. They considered that despite most municipalities have a mammogram from the Brazilian Unified Health System (SUS) in a 60 km radius, patients must travel this distance at least 4 times to undergo examination, going where the equipment is and back, and then once again to obtain the results of the exam[2]. In fact, this sum up to 240 km in travels to have exam access, and the quality of roads, travel costs or available means of transportation are not addressed as other possible aspects.
This inequity is augmented due to the Brazilian current opportunistic breast cancer screening policy, which depends on women to look for health care units on their own[3]. This means that they must find the time, usually during business days, and means of transportation all based in resources, knowledge and decision that prioritize early diagnosis of an unmanifested problem is at least as important as deal with more pressing ones like other present diseases, poverty or even the bills to pay. This makes that women with better access and that already have been examined before more inclined to screening programs[4]. All that can be associated with better education and higher income, both aspects that are more present in Brazilian state capitals than in other cities[5–7].
There are other possible epidemiologic explanations. For the last 40 years Brazil has been watching two know phenomena: Demographic Transition and Epidemiological transition. The Demographic transition is the change in the populational distribution towards a higher proportion of elderly people and populational concentration in urban areas. Improvements in living standards, for example in sanitation, education and health care, lead to decreasing fertility rates, lower birth rates and lower mortality rates in Brazil. That made the people live longer and changed the epidemiologic profile. This change is known as Epidemiological transition, that is the change in morbidity and mortality from predominantly infectious diseases to non-communicable diseases[8]. Cancer is among the diseases that had growing importance in the last years in Brazil, with breast cancer being acknowledged as main cause of death in women among malignant cancers. The epidemiological and demographic changes are ongoing processes that are spreading through Brazil, noticeably getting to the rural cites much later than states’ capitals[9].
Here are some other questions and suggestions:
In line 32 the authors say: “One method available is age-period-cohort studies, which seek to differentiate these elements in the… “. I believe that a method should not be used because it is available, but because it is the most suitable method to achieve a certain objective. Maybe replace it with: “The age-period-cohort method allows to differentiate these elements in the…”
The original idea was to imply that there are another possible methods. This is a good suggestion since it would emphasize better the choice for age-period-cohort method.
On line 95: delete “municipalities of the North, Northeast and Central-West regions.”
We accept this suggestion.
Line 109: It seems to me that in the Central-West region the best model was the one with 3 variables
We do agree that in Central-West region the best model is the one with 3 variables. This was part of a previous version of the original material of this research, where a p value of 0.01 was being considered.
Figure 1: Place title on y-axis.
The titles were added.
Table 1:
- Adjust the width of rows and columns to improve visualization
The rows and columns were adjusted.
- It is necessary to explain which test the p value refers to. I believe it is the test (F or Chi-square?) that compares the model in a given row with the model in the previous row.
The Chi square p value is now mentioned in the text and below the table.
Line 130: delete “by period”
We accept this suggestion.
Line 132: replace the expression “reference period” with the year in question.
We accept this suggestion.
Line 133: This expression is confusing: “surpassing the value of the reference period and continuing to increase until the end of the observed period, which occurs a little later for Brazil “
We have rewritten this part of the paragraph.
I believe that the following paragraph could be deleted (line 183), as it presents questions about the age-period-cohort models that have already been discussed in methodological articles and are not part of the questions of this study: “The age-period-cohort method has been used for many years and can provide ... requires the use of techniques such as zero-sum restriction to one of the parameters for each model.”
We accept this suggestion.
Line 193 to 200: The text is confusing and unfortunately the example does not help to understand the question.
The whole paragraph has been deleted, since the idea is not central to the objectives of this research.
Line 217: It is not clear how “changes in the formation of the breast epithelium, specific increases in high-grade cancers and late cancer diagnosis” can characterize a cohort effect.
That part has been rewritten to make the idea clearer.
Line 220: The authors say: “In general, the cohort effect indicates changes in the mortality rate from breast cancer in all age groups, in addition to the youngest group, with some that deserve attention”. Changes in the mortality rate in all age groups do not characterize a period effect?
Yes, it does. The sentence was corrected, and the paragraph rewritten.
Submission Date
13 June 2023
Date of this review
06 Jul 2023 02:29:14

Reviewer 2 Report
The current manuscript aims to analyze the effects of age, period, and birth cohort on the breast cancer mortality rate for Brazilian women, comparing state capitals and non-capital municipalities from 1980 to 2019. The study provides information that may be useful to guide country level policies to improve mortality trends or breast cancer. However, I have some concerns about the potential impact of the quality of the information for the study variables on the results reported by the authors. Some approaches made by the authors to deal with the quality of both census information and mortality data should be better justified since the study results and conclusions rely on that.
In this line, I provided some specific comments in order to underline different issues to improve.
Methods
Census information varied across the study period. The potential impact of different approaches over the study period (count, intercensus projections, population estimates, etc…) should be more developed in the methods section and the potential impact on the study results should be described in the discussion. Are there differences on the quality of the information by state capitals and non-capital municipalities?
I also have some concerns about the quality of the mortality data. The correction purposed using “ill-defined causes” should be better detailed in the methods section, not only by referencing another article.
Results
Across the abstract and the manuscript, the authors present their results as “high mortality”, “increased mortality in some regions”, etc.. but is it also important to provide specific rates and RR to the readership to interpret the results. I suggest adding a paragraph summarizing the overall (or the most relevant) mortality rates found by study groups. This information will also be very informative when comparing the results with other studies.
Discussion
Although the authors mentioned “the quality of the mortality data” as a study limitation, this section should be improved: in which direction the results may be biased by underreported deaths? Are they differences on the quality of mortality data by Brazilian regions (urban/rural area, etc)?. This is an important point to clarify since the study results may be directly affected.
Some of the results found, specially the cohort effect, may be explained by quality of information on mortality data?
Author Response
Dear reviewer 2,
Thank you for the feedback. Our answers and comments are highlighted in bold font.
Reviewer 2
Open Review
(x) I would not like to sign my review report
( ) I would like to sign my review report
Quality of English Language
(x) I am not qualified to assess the quality of English in this paper
( ) English very difficult to understand/incomprehensible
( ) Extensive editing of English language required
( ) Moderate editing of English language required
( ) Minor editing of English language required
( ) English language fine. No issues detected
|
Yes |
Can be improved |
Must be improved |
Not applicable |
|
|
Does the introduction provide sufficient background and include all relevant references? |
(x) |
( ) |
( ) |
( ) |
|
Are all the cited references relevant to the research? |
(x) |
( ) |
( ) |
( ) |
|
Is the research design appropriate? |
(x) |
( ) |
( ) |
( ) |
|
Are the methods adequately described? |
( ) |
( ) |
(x) |
( ) |
|
Are the results clearly presented? |
( ) |
( ) |
(x) |
( ) |
|
Are the conclusions supported by the results? |
( ) |
(x) |
( ) |
( ) |
Comments and Suggestions for Authors
The current manuscript aims to analyze the effects of age, period, and birth cohort on the breast cancer mortality rate for Brazilian women, comparing state capitals and non-capital municipalities from 1980 to 2019. The study provides information that may be useful to guide country level policies to improve mortality trends or breast cancer. However, I have some concerns about the potential impact of the quality of the information for the study variables on the results reported by the authors. Some approaches made by the authors to deal with the quality of both census information and mortality data should be better justified since the study results and conclusions rely on that.
In this line, I provided some specific comments in order to underline different issues to improve.
Methods
Census information varied across the study period. The potential impact of different approaches over the study period (count, intercensus projections, population estimates, etc…) should be more developed in the methods section and the potential impact on the study results should be described in the discussion. Are there differences on the quality of the information by state capitals and non-capital municipalities?
The available data in the database of the Department of Informatics of the Brazilian Unified Health System (DATA-SUS - https://datasus.saude.gov.br/) had limitations: in one set it covered from 1979 to 2012 and the others only offered data since 2000. The former is based on censuses, populational count and intercensal estimates, that are officially used for formal calculations, such as in taxes distribution. The latter is based on estimates, due to the long period between census assessments and methodological improvements in the estimation methods from 2013. In order to make the best possible use of all mortality available data, we decided to combine both bases as to cover the whole studied period.
I also have some concerns about the quality of the mortality data. The correction purposed using “ill-defined causes” should be better detailed in the methods section, not only by referencing another article.
Information about the chosen method and its steps has been added to the manuscript.
Results
Across the abstract and the manuscript, the authors present their results as “high mortality”, “increased mortality in some regions”, etc.. but is it also important to provide specific rates and RR to the readership to interpret the results. I suggest adding a paragraph summarizing the overall (or the most relevant) mortality rates found by study groups. This information will also be very informative when comparing the results with other studies.
A paragraph based on the data is added to the text, as well as the values themselves are inserted in the appendix section.
Discussion
Although the authors mentioned “the quality of the mortality data” as a study limitation, this section should be improved: in which direction the results may be biased by underreported deaths? Are they differences on the quality of mortality data by Brazilian regions (urban/rural area, etc)?. This is an important point to clarify since the study results may be directly affected.
Some of the results found, specially the cohort effect, may be explained by quality of information on mortality data?
The section where this issue is described has been rewritten to better show the interference of this problem.
Yes, we acknowledge that quality of information on mortality may play a role in the observed cohort effects, especially in previous ones, and that its improvement may be part of the trend observed in this study.

Round 2
Reviewer 2 Report
The authors have addressed the issues pointed by the reviewers. In my opinion, the paper has improved in its current version